# Separation of Mandelic Acid by a Reactive Extraction Method Using Tertiary Amine in Different Organic Diluents

**DOI:** 10.3390/molecules27185986

**Published:** 2022-09-14

**Authors:** Barış Kiriş, Yavuz Selim Aşçı, Muhammad Zahoor, Syed Shams ul Hassan, Simona Bungau

**Affiliations:** 1Department of Material and Material Processing Technologies, Vocational School of Technical Science, Istanbul University-Cerrahpasa, 34500 Istanbul, Turkey; 2Department of Chemistry, Faculty of Science, Istanbul University, 34452 Istanbul, Turkey; 3Department of Biochemistry, University of Malakand, Chakdara 18800, Pakistan; 4Shanghai Key Laboratory for Molecular Engineering of Chiral Drugs, School of Pharmacy, Shanghai Jiao Tong University, Shanghai 200240, China; 5Department of Natural Product Chemistry, School of Pharmacy, Shanghai Jiao Tong University, Shanghai 200240, China; 6Department of Pharmacy, Faculty of Medicine and Pharmacy, University of Oradea, 410028 Oradea, Romania

**Keywords:** separation, mandelic acid, reactive extraction, tri-n-octylamine

## Abstract

Mandelic acid is a valuable chemical that is commonly used in the synthesis of various drugs, in antibacterial products, and as a skin care agent in cosmetics. As it is an important chemical, various methods are used to synthesize and extract this compound. However, the yields of the used processes is not significant. A dilute aqueous solution is obtained when using several production methods, such as a fermentation, etc. In this study, the reactive extraction of mandelic acid from aqueous solutions using tri-n-octylamine extractant at 298.15 K was investigated. Dimethyl phthalate (DMP), methyl isobutyl ketone (MIBK), 2-octanone, 1-octanol, n-pentane, octyl acetate, and toluene were used as diluents. The batch extraction results of the mandelic acid experiments were obtained for the development of a process design. Calculations of the loading factor (Z), distribution coefficient (D), and extraction efficiency (E%) were based on the experimental data. The highest separation yield was obtained as 98.13% for 0.458 mol.L^−1^ of tri-n-octylamine concentration in DMP. The overall extraction constants were analyzed for the complex of acid-amine by the Bizek approach, including K_11_, K_12_, and K_23_.

## 1. Introduction

Mandelic acid, also known as almond acid, is an aromatic alpha-hydroxycarboxylic acid derived from the hydrolysis of the extract of bitter almonds [1]. It is a raw material and synthetic intermediate used for the preparation of pharmaceutical compounds such as antibiotics, drugs, etc., in medicine [2]. In addition, mandelic acid is a beneficial compound for use in antibacterial, healthcare and skincare products in the cosmetic and chemical industries [3,4]. For many years, mandelic acid has been used for treating skin problems such as acne, sun damage, or photoaging in skincare products [5,6]. It is an influential agent for the treatment of wrinkles and irregular pigmentation, to the same extent as glycolic acid, which is commonly used in skincare [7]. Glycolic acid has a smaller molecular structure and penetrates the skin deeply [8]. However, mandelic acid provides a slow and uniform penetration of the skin and irritates it less than the glycolic acid [9]. Therefore, mandelic acid has become a preferred skin care agent in recent years, used preferentially over glycolic acid. The production of mandelic acid can be carried out using various methods, including its chemical synthesis from potassium cyanide/benzaldehyde with chloroform. In addition to chemical synthesis, it can also be produced via biotechnological processes, such as the conversion of benzoylformic acid into mandelic acid using the micro-organisms of micrococcus freudenreichii, micrococcus luteus, enterococcus faecalis, and enterococcus faecalis [10,11]. On the other hand, mandelic acid can be derived from the extraction of almonds with diluted hydrochloric acid [10]. By biotechnological processes and extraction methods, diluted aqueous solutions of mandelic acid are obtained. Therefore, the separation of mandelic acid from aqueous solutions and its conversion into commercial forms occur as significant processes.

A considerable number of studies have been published on the separation of carboxylic acids from aqueous solutions by ion-exchange chromatography [12], adsorption [13], electro dialysis [14], ultrafiltration [14], anion exchange [15], liquid−liquid extraction [16], membrane separation [17], reverse osmosis [18], and precipitation [19]. Recently, among these methods, the reactive extraction method, which requires less energy input and leads to a high selectivity and efficiency, has gained attention [20]. Many researchers have studied the reactive extraction by using several extractants, including tri-n-butylamine (TBA), tri-n-butyl phosphate (TBP), tri-n-octyl phosphine oxide (TOPO), tri-n-propylamine (TPA), Alamine 336, Aliquat 336, and Amberlite LA-2. Table 1 offers a brief review of the reactive extractions of various carboxylic acids using different types of extractants. Tri-n-octylamine (TOA), a tertiary amine compound [21,22,23,24,25,26], has been widely used as an effective reagent for the reactive extraction of various carboxylic acids, such as lactic acid [27], malic acid [28], propionic acid [29], and succinic acid [30]. Caşcaval et al. [31] studied the reactive extraction of acetic acid using tri-n-octylamine dissolved in dichloromethane, butyl acetate, and n-heptane. Reactive extraction experiments and modeling studies of citric acid using tri-n-butyl phosphate, tri-n-octylamine, and Aliquat 336 were performed by Thakre et al. [32] Datta et al. [33] carried out extraction experiments using glycolic acid from aqueous solutions with tri-n-octylamine and tridodecylamine. As seen from the literature, the reactive extraction of mandelic acid using tri-n-octylamine has not yet been performed.

In this work, we studied the reactive extraction of mandelic acid from aqueous phase by tri-n-octylamine using different diluents. For this purpose, an alcohol (1-octanol), two different ketones (2-octanone and methyl isobutyl ketone), two different esters (dimethyl phthalate and octyl acetate), aromatic hydrocarbon (toluene), and alkane (n-pentane) were chosen as diluents.

## 2. Results and Discussion

The reaction mechanism of mandelic acid (MA) with tri-n-octylamine (TOA) can be described by Equation (1):(1)$${\mathrm{mMA}}^{\mathrm{aq}}+{\mathrm{nTOA}}^{\mathrm{org}}={\left[{\left(\mathrm{MA}\right)}_{m}.\;{\left(\mathrm{TOA}\right)}_{n}\right]}^{\mathrm{org}}$$

Herein, the m moles of undissociated mandelic acid react with n moles of tri-n-octylamine at the external interface between the aqueous phase (aq) and the organic phase (org) to form 1 mol of the complex [(MA)_m_.(TOA)_n_]. This equation can be demonstrated by the overall thermodynamic constant [28]:(2)$$K_{m,n}{}^{\mathrm{org}}={\left[{\left(\mathrm{MA}\right)}_{m}.{\left(\mathrm{TOA}\right)}_{n}\right]}^{\mathrm{org}}\;/\;{\left({\left[\mathrm{MA}\right]}^{m}\right)}^{\mathrm{aq}}\;{\left({\left[\mathrm{TOA}\right]}^{n}\right)}^{\mathrm{org}}$$

The loading factor (Z) is obtained by dividing the total amount of mandelic acid in the organic phase C_ma,org_ by the total amount of tri-n-octylamine in the organic phase C_TOA,org_. This expression can be written as follows [28]:(3)$$Z={\;C}_{\mathrm{ma},\mathrm{org}}/C_{\mathrm{TOA},\mathrm{org}}$$

The distribution coefficients (D) of the mandelic acid extracted from the aqueous phase transitioning into the organic phase and the efficiency of extraction (E) can be calculated by Equations (4) and (5), respectively [29]:(4)$$D={\;C}_{\mathrm{ma},\mathrm{org}}/C_{\mathrm{ma}}$$
(5)$$E=\left[1-(C_{\mathrm{ma}}/C_{\mathrm{ma}0})\right]\;.100$$

In Equation (5), C_ma_ is the concentration of the mandelic acid in the aqueous phase after extraction and C_ma0_ is the initial concentration of mandelic acid in the aqueous phase. An efficiency of extraction of 100% means that all of the mandelic acid in the aqueous phase has been removed.

The results of the reactive extraction experiments are displayed in Table 2. Additionally, the conventional extraction results without tri-n-octylamine are presented in the same table. The initial concentration of mandelic acid was 0.74 mol.L^−1^ (10% *w*/*w*) in the aqueous phase, and the concentrations of tri-n-octylamine in the diluents ranged from 0.092 mol.L^−1^ and 0.458 mol.L^−1^ in both experiments. 

The results showed that the conventional extractions were realized with a low level of efficiency, and the distribution coefficients ranged from 0.01 to 0.78 without tri-n-octylamine in the n-pentane, octyl acetate, and toluene. It was observed that more than 70% of the mandelic acid could be separated with the other diluents, DMP, MIBK, 2-octanone, and 1-octanol. This can be attributed to the polarity of these diluents, which gives them a high extraction efficiency. However, the polarity alone is not sufficient to completely explain the solvating ability. In this study, the use of an alcohol, such as 1-octanol, which has a high hydrogen binding capacity, led to high distribution coefficients [33,61].

To achieve higher yields, the reactive extraction experiments were also performed, and the results indicated that the extraction efficiency and the dispersion coefficients increased with an increasing tri-n-octylamine concentration in all the diluents. The high extraction efficiency was found to be 98.13% using DMP with 0.458 mol.L^−1^ of tri-n-octylamine concentration. It was seen that the E% values ranged between 30.67 and 98.13 with increasing concentrations of tri-n-octylamine from 0.092 to 0.458 mol.L^−1^. Figure 1 shows the plot of the E% values for all the diluents employed.

The distribution coefficients, which ranged from 0.44 to 52.5 with increasing of tri-n-octylamine concentrations, were calculated using Equation (4). The tri-n-octylamine concentration vs. distribution coefficient for each diluent is presented in Figure 2.

Figure 3 displays the concentration of tri-n-octylamine for each diluent against the loading factor. The loading factors had high values, ranging from 1.53 to 6.99, and it can be stated that the system was an overloading one. In this case, the complex of [(MA)_m_.(TOA)_n_] was formed with more than one mandelic acid per tri-n-octylamine [61]. 

The increase in the basic amine compound concentration could lead to a decrease in the polarity and dissolution. As can be seen from Figure 3, the values of Z decreased with the increasing tri-n-octylamine concentration. In the literature, the decrease in the loading factors has been described as an indication of the reaction mechanism, in which the complexes contain more than one amine per complex [33,61].

The Bizek approach is commonly used for predicting complex formations in reactive extractions [62]. By this approach, the stoichiometry of the complexes can be formed as (MA).(TOA), (MA).(TOA)_2_ and (MA)_2_.(TOA)_3_. K_11_, K_12_, and K_23_, which were the overall extraction constants, were analyzed using the following equations, respectively:(6)$$K_{11}\;;{\left(\mathrm{MA}\right)}^{\mathrm{aq}}+{\left(\mathrm{TOA}\right)}^{\mathrm{org}}↔{\left[\left(\mathrm{MA}\right).\;\left(\mathrm{TOA}\right)\right]}^{\mathrm{org}}$$
(7)$$K_{12}\;;{\left(\mathrm{MA}\right)}^{\mathrm{aq}}+2{\left(\mathrm{TOA}\right)}^{\mathrm{org}}↔{\left[\left(\mathrm{MA}\right).\;{\left(\mathrm{TOA}\right)}_{2}\right]}^{\mathrm{org}}$$
(8)$$K_{23}\;;2{\left(\mathrm{MA}\right)}^{\mathrm{aq}}+3{\left(\mathrm{TOA}\right)}^{\mathrm{org}}↔{\left[{\left(\mathrm{MA}\right)}_{2}.{\left(\mathrm{TOA}\right)}_{3}\right]}^{\mathrm{org}}$$

Table 3 shows the overall extraction constants calculated using Equations (6)–(8). K_11_ was calculated for all the solvents. According to the Bizek approach, we calculated K_23_ for the non-protonating diluents DMP, MIBK, 2-octanone, n-pentane, octyl acetate, and toluene, and only K_12_ for 1-octanol.

## 3. Materials and Methods

Mandelic acid (>99%; Table 4) and tri-n-octylamine (for synthesis, >98%) were purchased from Sigma-Aldrich and Merck, respectively. Dimethyl phthalate (>99%, Merck), methyl isobutyl ketone (>99%, Merck), 2-octanone (>99%, Carlo Erba), 1-octanol (>99%, Merck), n-pentane (>98%, Sigma-Aldrich), octyl acetate (>98%, Sigma-Aldrich), and toluene (>99%, Carlo Erba) were used as diluents without further purification. Table 2 shows some of the physical and chemical properties of mandelic acid [10]. 

Two different phases, the aqueous and organic, were prepared as follows. The appropriate amounts of mandelic acid were dissolved in distilled water (aqueous phase) to prepare the initial acid concentrations of 0.74 mol.L^−1^ (~10% by weight). For the organic phase, tri-n-octylamine was dissolved in the diluents with constant concentrations of 0.092 mol.L^−1^, 0.183 mol.L^−1^, 0.275 mol.L^−1^, 0.366 mol.L^−1^, 0.458 mol.L^−1^. A total of 5 mL of the aqueous phase and 5 mL of the organic phase were mixed in Erlenmeyer flasks and balanced on a Nuve Shaker-ST402 at 298.15 K for 4 h. The pre-tests showed that a time of 4 h was optimal for balancing the reactive extractions. All samples were centrifuged at 2000 rpm for 10 min by Nuve CN 180 to allow for the complete separation of the phases.

SCHOTT TitroLine^®^ Easy M2 was used to analyze the concentration of mandelic acid in the aqueous phase with sodium hydroxide (relative uncertainty: ±1%). Each analysis was performed three times to minimize the errors. Experimental data were used to calculate the loading factor (Z), distribution coefficient (D), and extraction efficiency (E%).

## 4. Conclusions

The extraction of mandelic acid from aqueous phases by tri-n-octylamine in seven diluents at 298.15 K was investigated. As a result, the extraction efficiency of mandelic acid with tri-n-octylamine was found to be high, especially in the case of polar diluents such as DMP, 2-octanone, and MIBK. The maximum extractability of mandelic acid was 98.13% with DMP (0.458 mol.L^−1^ concentration of tri-n-octylamine). The maximum extraction efficiencies for diluents used with maximum concentration of tri-n-octylamine were identified as DMP > 2-octanone > MIBK > toluene = octyl acetate > 1-octanol > n-pentane. Using Bizek approach, the overall extraction constants of K_11_, K_12_, and K_23_ were calculated as 4.81–114.63 mol.L^−1^, 123.02–620.05 L^2^.mol^−2^, and 9.37–8286.93 L^4^.mol^−4^, respectively. These results show that the reactive extraction method is an effective method for the separation of mandelic acid, and tri-n-octylamine is a compatible reagent for this separation process.

## Figures and Tables

**Figure 1 molecules-27-05986-f001:**
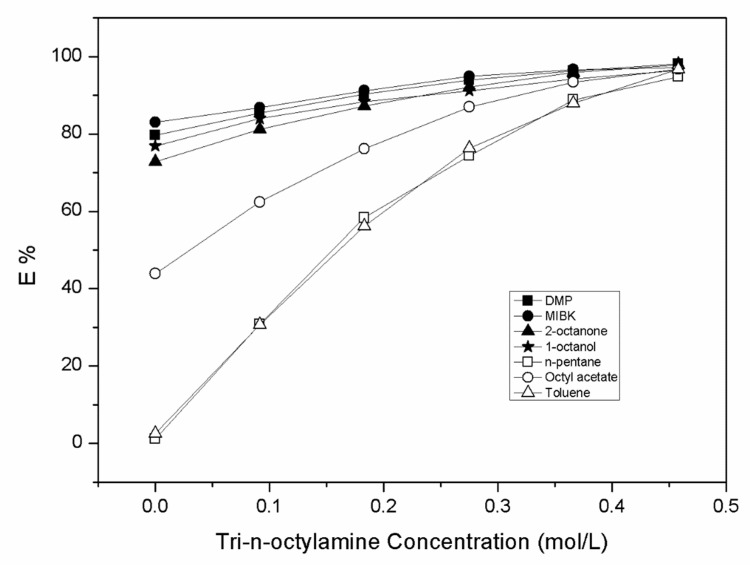
Plot of extraction efficiency (E) versus the molar concentration of TOA (C_TOA,org_) in different diluents.

**Figure 2 molecules-27-05986-f002:**
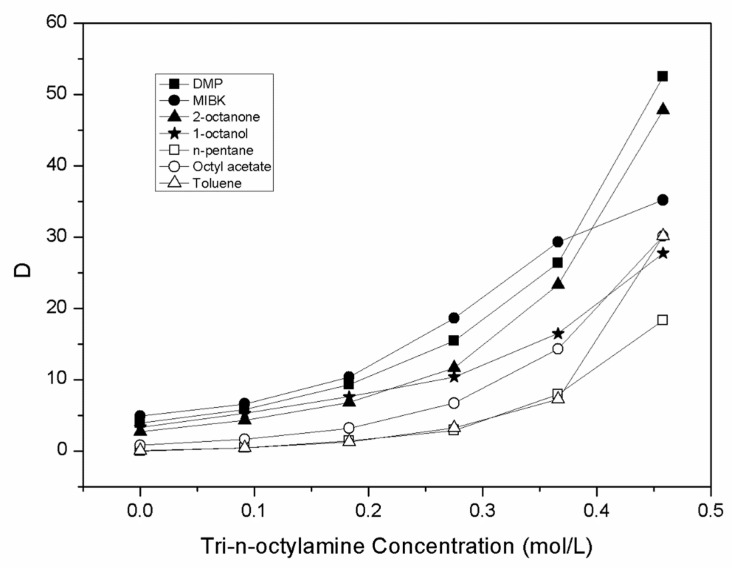
Plot of the distribution coefficients (D) versus the molar concentrations of TOA (C_TOA,org_) in different diluents.

**Figure 3 molecules-27-05986-f003:**
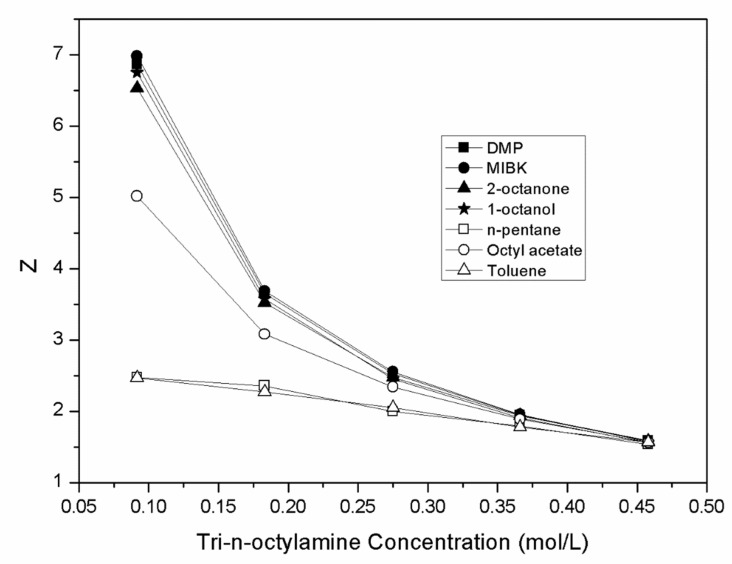
Plot of the loading factors (Z) versus the molar concentrations of TOA (C_TOA,org_) in different diluents.

**Table 1 molecules-27-05986-t001:** A brief review of the reactive extraction using various extractants.

Type of Extractant	Type of Carboxylic Acid	Ref.
Alamine 336	Acetic acid, lactic acid, succinic acid, malonic acid, fumaric acid, maleic acid	[34]
	Pyruvic acid	[35]
Aliquat 336	Propionic acid	[36]
Amberlite LA-2	Acrylic acid	[37]
	Citric acid	[38]
	Gibberellic acid	[39]
	Glycolic acid	[40]
	Malic acid	[41]
	Nicotinic acid	[42]
	Oxoethanoic acid	[43]
	Pentanedioic acid	[44]
	Picric acid	[45]
	Rosmarinic acid	[46]
	Succinic acid	[47]
	Tartaric acid	[48]
Tri-n-butylamine	Acetic acid	[16]
Tri-n-butyl phosphate	Propionic acid	[49]
	Butyric acid, lactic acid, tartaric acid, itaconic acid, succinic acid, citric acid	[50]
	Lactic acid	[51]
Tri-n-octylamine	Acetic acid	[22]
	Citric acid	[52]
	Glutaric acid	[53]
	Glycolic acid	[33]
	Lactic acid	[54]
	Malic acid	[28]
	Nicotinic acid	[55]
	Picolinic acid	[56]
	Propionic acid	[29]
	Pyruvic acid	[57]
	Succinic acid	[26]
Tri-n-octylphosphine oxide	Nicotinic acid	[58]
	Propionic acid	[59]
Tri-n-propylamine	Acetic acid	[60]

**Table 2 molecules-27-05986-t002:** Molar concentrations of mandelic acid in water/organic diluents and variations in the loading factor (Z), distribution coefficient (D), and extraction efficiency (E) at 298.15 K.

Diluent	C_TOA,org_ (mol.L^−1^)	C_ma_ (mol.L^−1^)	C_ma,org_ (mol.L^−1^)	D	Z	E (%)
DMP	0.000	0.151	0.589	3.91	-	79.65
	0.092	0.108	0.632	5.84	6.87	85.37
	0.183	0.072	0.668	9.31	3.65	90.30
	0.275	0.045	0.695	15.44	2.53	93.92
	0.366	0.027	0.713	2635	1.95	96.34
	0.458	0.014	0.726	52.50	1.59	98.13
MIBK	0.000	0.125	0.615	4.91		83.07
	0.092	0.097	0.643	6.60	6.99	86.84
	0.183	0.065	0.675	10.38	3.69	91.21
	0.275	0.038	0.702	18.64	2.55	94.91
	0.366	0.024	0.716	29.32	1.96	96.70
	0.458	0.020	0.720	35.20	1.57	97.24
2-Octanone	0.000	0.201	0.539	2.68		72.81
	0.092	0.139	0.601	4.33	6.53	81.23
	0.183	0.095	0.645	6.82	3.53	87.21
	0.275	0.058	0.682	11.68	2.48	92.11
	0.366	0.030	0.710	23.36	1.94	95.90
	0.458	0.015	0.725	47.84	1.58	97.95
1-Octanol	0.000	0.171	0.569	3.34		76.95
	0.092	0.118	0.622	5.25	6.76	84.00
	0.183	0.086	0.654	7.61	3.57	88.39
	0.275	0.065	0.675	10.38	2.45	91.21
	0.366	0.042	0.698	16.48	1.91	94.28
	0.458	0.026	0.714	27.76	1.56	96.52
n-Pentane	0.000	0.732	0.008	0.01		1.09
	0.092	0.512	0.228	0.44	2.48	30.78
	0.183	0.308	0.432	1.40	2.36	58.34
	0.275	0.190	0.550	2.89	2.00	74.32
	0.366	0.083	0.657	7.97	1.80	88.85
	0.458	0.038	0.702	18.30	1.53	94.82
Octyl acetate	0.000	0.415	0.325	0.78		43.91
	0.092	0.278	0.462	1.66	5.02	62.41
	0.183	0.176	0.564	3.20	3.08	76.20
	0.275	0.096	0.644	6.71	2.34	87.02
	0.366	0.048	0.692	14.31	1.89	93.47
	0.458	0.024	0.716	30.16	1.56	96.79
Toluene	0.000	0.721	0.019	0.03		2.51
	0.092	0.513	0.227	0.44	2.47	30.67
	0.183	0.324	0.416	1.28	2.27	56.19
	0.275	0.175	0.565	3.22	2.05	76.29
	0.366	0.089	0.651	7.29	1.78	87.94
	0.458	0.024	0.716	30.16	1.56	96.79

**Table 3 molecules-27-05986-t003:** The overall extraction constant values of K_11_, K_12_, and K_23_ in different diluents.

Diluent	C_TOA,org_ (mol.L^−1^)	K_11_ (mol.L^−1^)	K_12_ (L^2^.mol^−2^)	K_23_ (L^4^.mol^−4^)
DMP	0.092	63.45		586.19
	0.183	50.88		709.07
	0.275	56.16		1247.94
	0.366	71.99		2660.25
	0.458	114.63		8286.93
MIBK	0.092	71.72		736.49
	0.183	56.70		871.57
	0.275	67.78		1798.76
	0.366	80.10		3281.31
	0.458	76.87		3760.70
2-Octanone	0.092	47.02		338.46
	0.183	37.25		393.38
	0.275	42.48		727.90
	0.366	63.83		2101.43
	0.458	104.45		6893.06
1-Octanol	0.092	57.04	620.05	
	0.183	41.61	227.38	
	0.275	37.73	137.20	
	0.366	45.02	123.02	
	0.458	60.60	132.32	
*n*-Pentane	0.092	4.83		9.43
	0.183	7.65		24.82
	0.275	10.52		55.38
	0.366	21.77		263.76
	0.458	39.95		1041.92
Octyl acetate	0.092	18.05		64.89
	0.183	17.49		99.30
	0.275	24.38		253.92
	0.366	39.10		808.89
	0.458	65.85		2773.00
Toluene	0.092	4.81		9.37
	0.183	7.01		21.62
	0.275	11.70		66.68
	0.366	19.92		223.09
	0.458	65.85		2773.00

**Table 4 molecules-27-05986-t004:** Properties of mandelic acid.

Molecular Formula	C_6_H_5_CH(OH)COOH
Molecular weight	152.147 g/mol
Appearance	White crystalline solid
Density	1.3 g/cm^3^
Melting point	131–135 °C
Solubility (in water)	158.7 g/cm^3^

## Data Availability

Not applicable.

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
