# Peer review of "Separation of Mandelic Acid by a Reactive Extraction Method Using Tertiary Amine in Different Organic Diluents"

_molecules, 2022, doi:10.3390/molecules27185986_

Round 1

Reviewer 1 Report

manuscript is well prepared and can be published

Author Response

Thank you worthy reviewer for the positive input

Reviewer 2 Report

This paper deals with the extraction of mandelic acid with tri-octylamine.

The knowledge of the previous papers was just applied to this extraction system. Reviewer consider Following points should be considered.

1. In Introduction, so far there are many studies on the extraction of organic acids. References are biased toward author's paper. 

2. From Fig. 1, in the absence of TOA the extractability is high enough. Why is the reactive extraction system applied to this extraction?

3. In Fig. 2, the calculated lines should be presented.

Author Response

Reviewer 2

This paper deals with the extraction of mandelic acid with tri-octylamine.

The knowledge of the previous papers was just applied to this extraction system. Reviewer consider Following points should be considered.

  • Thank you, worthy reviewer for your positive input.

  1. In Introduction, so far there are many studies on the extraction of organic acids. References are biased toward author's paper. 
  • Worthy reviewer, the references were replaced with relevant one. Hope it will be ok now.

  1. From Fig. 1, in the absence of TOA the extractability is high enough. Why is the reactive extraction system applied to this extraction?
  • TOA was used as a reagent in the extraction process. In addition, 7 different solvents with different chemical structures were used. As, these solvents differ from each other in properties such as volatility, toxicity etc which definitely would have different effects. When the results were examined, although some solvents were able to provide high extraction efficiency in physical extraction studies without using TOA, a successful separation could not be achieved, especially toluene, octyl acetate and n-heptane. However, with the addition of TOA, these solvents reached to a high yields as the others as can be seen at Fig. 1. This is important because it shows that the use of TOA allows to achieve good yields.

  1. In Fig. 2, the calculated lines should be presented.
  • On your suggestion, using the calculated lines method, Fig. 2 has been drawn. However, since the results of seven different solvents are presented in the graph, there has been a lot of confusion in the recommended method visually. Therefore, the chart is left as it is.

Reviewer 3 Report

In the article of Kiriş et al. entitled „ Separation of mandelic acid by reactive extraction method using tertiary amine in different organic diluents” Authors developed an extraction method of mandelic acid from aqueous solutions using f tri-n-octylamine dissolved in seven different organic solvents. The highest extraction efficiency was obtained with 0.458M solution of tri-n-octylamine in dimethyl phthalate

The overall impression is that the paper is well written, with proper use of citations and in my opinion is suitable for publication in Molecules in its current form.

Author Response

Thank you for the positive input. 

Round 2

Reviewer 2 Report

As described in the previous review, the knowledge of the previous papers was just applied to this extraction system and  there are no academic and practical finding. This point  is not improved.